# Recent Progress of Electrode Architecture for MXene/MoS_2_ Supercapacitor: Preparation Methods and Characterizations

**DOI:** 10.3390/mi13111837

**Published:** 2022-10-27

**Authors:** Muhammad Akmal Kosnan, Mohd Asyadi Azam, Nur Ezyanie Safie, Rose Farahiyan Munawar, Akito Takasaki

**Affiliations:** 1Fakulti Kejuruteraan Pembuatan, Universiti Teknikal Malaysia Melaka, Hang Tuah Jaya, Durian Tunggal, Melaka 76100, Malaysia; 2Department of Engineering Science and Mechanics, Shibaura Institute of Technology, 3 Chome-7-5 Toyosu, Koto City, Tokyo 135-8548, Japan

**Keywords:** MXene, MoS_2_, hybrid electrode compositions, characterization, supercapacitor, specific capacitance

## Abstract

Since their discovery, MXenes have conferred various intriguing features because of their distinctive structures. Focus has been placed on using MXenes in electrochemical energy storage including a supercapacitor showing significant and promising development. However, like other 2D materials, MXene layers unavoidably experience stacking agglomeration because of its great van der Waals forces, which causes a significant loss of electrochemically active sites. With the help of MoS_2_, a better MXene-based electrodecan is planned to fabricate supercapacitors with the remarkable electrochemical performance. The synthesis of MXene/MoS_2_ and the ground effects of supercapacitors are currently being analysed by many researchers internationally. The performance of commercial supercapacitors might be improved via electrode architecture. This analysis will support the design of MXene and MoS_2_ hybrid electrodes for highly effective supercapacitors. Improved electrode capacitance, voltage window and energy density are discussed in this literature study. With a focus on the most recent electrochemical performance of both MXene and MoS_2_-based electrodes and devices, this review summarises recent developments in materials synthesis and its characterisation. It also helps to identify the difficulties and fresh possibilities MXenes MoS_2_ and its hybrid heterostructure in this developing field of energy storage. Future choices for constructing supercapacitors will benefit from this review. This review examines the newest developments in MXene/MoS_2_ supercapacitors, primarily focusing on compiling literature from 2017 through 2022. This review also presents an overview of the design (structures), recent developments, and challenges of the emerging electrode materials, with thoughts on how well such materials function electrochemically in supercapacitors.

## 1. Introduction

Energy storage systems have become crucial in electronic, electrical, defense, and locomotive technologies due to the current scenario of obtaining energy from clean, sustainable and renewable sources [1,2]. Due to their quick power delivery, outstanding rate performance, and long cycling life, supercapacitors have gained more and more attention as an example of electrochemical energy-storage devices [3,4]. Other 2D materials, such as transition metal dichalcogenides (TMDs), hexagonal boron nitrides, metal oxides, black P, silicene, germanene, and MXenes, have all received a great deal of increased interest since graphene [5] was awarded the Nobel Prize in 2010 [6,7,8,9,10,11,12,13]. Since 2011’s discovery of Ti_3_C_2_, research on 2D transition metal carbides, nitrides, and carbonitrides, which are referred to collectively as MXenes, is one that grows at a rapid pace. Flexible and wearable technologies have been extensively researched in recent decades due to their prospective uses in portable mobile electronics and human motion tracking. MXene, a novel developing family of 2D nanomaterials, has superiorities such as excellent electrical conductivity, abundant terminal groups, a unique layered-structure, a high surface area, and hydrophilicity, making it a possible candidate material for flexible and wearable electronics. Numerous pioneering works have been dedicated to the development of flexible MXene-based composites with varied functionalities and tailored structures [14]. Because of its excellent hydrophilic surfaces, great chemical stability, variable interlayer spacing, and high electrical conductivity, Ti_3_C_2_T_x_, a typical example of a 2D MXene, has been the subject of extensive research material for supercapacitors and batteries [15]. Ti_3_C_2_ MXene has great cycling and superior rate performance; nevertheless, its applications are restricted due to its low specific capacitance. Currently, several different experiments are being conducted to enhance the specific capacitance of Ti_3_C_2_ MXene.

A single stack of MXenes has a thickness of around 1 nm and is constructed from atomic layers. There are reports that more than 20 different MXene compositions have been synthesised, and over 70 possible compositions of MXenes are theoretically anticipated. Different combinations of M and X can produce different compositions of MXenes [16]. A 2D flake of MXene has a general formula of M_n+1_X_n_T_x_ and contains n + 1 (n = 13) layers of early transition metals interspersed with n layers of carbon or nitrogen. The surface terminations that are attached to the outer M layers, such as O, OH, F, and/or Cl, are denoted by the symbol Tx in the formula as referred to in Figure 1. MXenes come in a wide range of compositions and topologies, which has resulted in the development of a sizable and quickly growing family of 2D materials [17]. At the bottom of Figure 1, three different types of MXenes’ atomic schematics are displayed. Unlike graphene and the majority of other 2D materials, MXene is a two-dimensional transition metal carbide, nitride, or carbonitride that exhibits extremely accessible hydrophilic surfaces. MXene sensors/biosensors provide good reproducibility of data over an extended period of time due to their wide surface area, strong biocompatibility, and long-term stability [18]. The MAX phase powder is mixed for a specific amount of time in aqueous hydrofluoric acid (HF), then centrifuged and repeatedly washed with distilled water until the pH level reaches between 4 and 6. This process produces a 2D MXene sheet. As a result, MXene, or loosely packed, exfoliated, graphite-layered structures, are created. 

By integrating flakes with good hydrophilicity and dispersion stability in an aqueous dispersion, the wet etching environment dissolved the aluminium layers and caused the surface of the MXene sheets to be terminated with functional groups such as =O, OH, and -F. However, MXenes experience substantial oxidative degradation, which severely impairs all of their properties and hinders future applications. In order to extend the shelf life of MXenes for practical applications, it is tough to improve their oxidation stability. Significant advancements in the quality-control and oxidation stability of MXenes have been made in the ten years from their discovery (between 2011 and 2020), indicating that each step from synthesis through storage and use has a significant influence on the characteristics of MXenes as a result [19]. Since water and oxygen molecules are the main contributors to MXene oxidation, Eunji et al. anticipate that the continuous ZIF-8 coating will effectively prevent them from entering the MXene surface.

For instance, the multilayer Ti_3_C_2_T_x_ powders are capable of delaminating in DMSO to create few-layer MXene flakes. These flakes have a significantly increased specific capacitance of 130 F g^−1^ with 1 m KOH as its electrolyte compared with Ti_3_C_2_ MXene, which only has 70 F g^−1^ in this electrolyte configuration [20]. Regarding heterostructures based on Ti_3_C_2_, however, several investigations have been conducted by putting together materials similar to batteries with Ti_3_C_2_ [21]. Ti_3_C_2_ MXene demonstrates exceptional performance as capacitive supercapacitor electrodes. Hence, creating 2D capacitive materials with Ti_3_C_2_ MXene is an intriguing prospect [22]. Table 1 depicts the overall structure as well as the chemical makeup of all functional group-free MXenes that have been described. Since the discovery of the first Mxene, Ti_3_C_2_T_x_, 20 and keep counting, distinct MXene compositions have been successfully created in the lab. X will either be C or N. Tx is a functional group created by etching, such as -OH or -F. For instance, (Mo_4_V) C4 is an MXene of the M_5×4_ type with n = 4. Mo and V are the two different transition element types present here, while X is carbon [23]. 

MXenes are typically placed onto different substrates or combined with other substances (polymers, inorganic substances, etc.) to create composite films. Another tactic is to modify MXene films’ internal structure to boost ion mobility and boost the energy storage capacity. In response to these issues, a number of novel techniques and studies on the enhancement of the electrochemical performance of MXene films emerged as shown in Table 1.

MoS_2_, one example of a typical 2D transition metal dichalcogenide (TMD) that is not excluded, has also been the subject of significant research. The molybdenum sulphide molecule is made up of S-Mo-S atoms that are covalently bound to one another and are only held together by the meagre van der Waals forces that it has [24,25]. Two-dimensional MoS_2_ has attracted significant interest due to its potential application as a semiconducting material with a direct bandgap. MoS_2_ is a promising material for applications such as hydrogen storage, gas sensing at room temperature, lubricants, coatings for space applications, coatings against aluminium alloys, catalysts, batteries, and nano-transistors [26,27,28,29,30,31,32,33,34,35,36]. It is common knowledge that MoS_2_ can exist in many distinct polymorphs. The two main crystal structures of MoS_2_ are the metastable 1T phase (1T-MoS_2_) and the stable 2H phase (2H-MoS_2_) with a monolayer bandgap of 1.9 eV that causes it to behave like a semiconductor). It has been demonstrated that 1T-MoS_2_ is a suitable material for the electrodes of supercapacitors with high capacitance [37].

Compared to graphene, the conductivity of MoS_2_ and dielectric constant is lower. As a result, it has the ability to provide superior impedance matching with open space. It may effectively boost the impedance matching when combined with MXene, which has high conductivity, and it can also lower the surface reflection generated by the high conductivity. 

MXenes are fundamentally depending on the technique utilized during synthesis, the method used to produce the electrodes, and the method used to construct the MXene. To summarise, the etching procedure and subsequent processing environment have a considerable impact on the phase composition and microstructure of MXenes, as well as the associated physical/chemical properties. The more advanced synthesis procedures lay the groundwork for their widespread use. The adjustable microstructure and rich surface chemistry are also crucial in determining MXene characteristics. At low temperatures, the dark condition, vacuum, and inert environment are excellent methods of isolating moisture and oxygen and preventing oxidation behaviour, particularly for few-layered MXenes. This oxidation action is common at high temperatures, and associated transition metal oxides are generated. The phase transition temperature and products are typically determined by MXene content and atmospheric conditions. Morphology and structure are also crucial aspects to consider. Electrochemical characteristics have been improved thanks to hierarchical structures of MXenes together with hybrid variations of MXenes with carbon Nanotubes (CNTs), MoS_2_, transition metal oxides, and others [38]. In order to achieve the highest possible output, it will be helpful to fabricate MXene anode sheets using suitable chemicals. The MoS_2_/MXene electrode exhibits a redox profile in an expanded voltage window up to about 3.0 V with aqueous electrolytes by amalgamating the benefits of MoS_2_ and MXene. Aqueous electrolytes typically exhibit limited cell voltage [39], but organic electrolytes attain an extended working voltage window [40]. Although the combination of MoS_2_ and MXene enhance the total capacitance and the cell voltage, the improvement in the operating voltage window may also be influenced by the electrode materials. This condition makes MoS_2_/MXene a promising electrode material for developing supercapacitors of the latest iteration that have a high energy density. This review summarises recent developments in materials synthesis, basic properties, and characteristics. It also helps to identify the difficulties and fresh possibilities MXenes, MoS_2_, and its hybrid heterostructure in this developing field of energy storage. Following the review, readers will understand the various methodologies for synthesizing MXenes/MoS_2_ better, as well as the influence of the electrochemical performance of MXenes/MoS_2_ on their shape and surface chemistry. In applications involving supercapacitors, there is also an emphasis placed on various anode architectures.

## 2. Overview of the Methods Used to Synthesize MXene/MoS_2_

The idea behind synthesizing the Mxene/MoS_2_ hybrid heterostructure is fundamental, and the process flow is straightforward. To begin, scientists need to synthesize a single molecule of MXene and a single molecule of MoS_2_. After that, either of the materials can be combined with the other. Titanium-based MXenes (Ti_2_CT_x_ and Ti_3_C_2_T_x_) are one of the most popular chemicals produced and explored extensively through experimentation for years. There are both bottom-up and top-down methods for synthesising MXene, which are commonly used. Chemical exfoliation of the MAX phase that is available commercially is a typical method to acquire both of these things. 

Hui et al. in 2020 recorded that Ti_3_C_2_T_x_ has demonstrated a high volumetric capacitance of B1500 F cm^3^ (380 F g^−1^, 90 nm thickness) in the H_2_SO_4_ electrolyte, serving as a representative of MXenes. Even though the Ti_3_C_2_T_x_ MXene electrode sheets had a thickness of up to 200 micrometres, they nevertheless showed remarkable performance. Transparent solid-state supercapacitors can be made from the very transparent and conductive Ti_3_C_2_T_x_ sheets. The aforementioned findings show that they have promising potential for use in supercapacitors. Exfoliation of Ti_3_AlC_2_ is shown schematically in Figure 2, A of the Ti_3_AlC_2_ MAX structure; B, development of the -OH terminal following the etching of the aluminium; C, the formation of MXene sheets following the final exfoliation procedure to break hydrogen bonds. In most cases, the synthesis of MXene is broken down into three stages: (a) the creation of the Ti_3_AlC_2_ precursor MAX phase; (b) the etching off of the Al layer; and (c) the exfoliation and intercalation of the MXene. For instance, HF, LiF-HCl, and a select few acids that do not contain fluoride, such as NH_4_Cl, are the primary etchants utilized in numerous works. The equation for the HF reaction with Ti_3_AlC_2_ is as follows: (1). The complimentary processes that result in the synthesis of fluorinated and hydroxinated Ti_3_C_2_ are represented by Equations (2) and (3), respectively [41]:Ti_3_AlC_2_ + 3HF → AlF_3_ + 1.5H_2_ + Ti_3_C_2_(1)
Ti_3_C_2_ + 2H_2_O → Ti_3_C_2_(2)
(OH)_2_ + H_2_Ti_3_C_2_ + 2HF → Ti_3_C_2_ F_2_ + H_2_(3)

The etching process has two most popular approaches: HF and in situ HF (Figure 3). In the HF method, researchers will employ three distinct concentrations of HF throughout three separate times, and this will be adequate to synthesise Ti_3_C_2_. After that, multilayer Ti_3_C_2_T_x_ will be produced from Ti_3_AlC_2_ by using a mixture of hydrochloric acid (HCl) and lithium fluoride (LiF) as the starting materials. Clay synthesis is another name for this process. When larger flake sizes, high electrical conductivity and improved mechanical qualities are all things that are sought, it is recommended that a LiF-HCl etchant be employed.

### 2.1. Preparation of Ti_3_C_2_ MXene

This review will be selective and focus on methods appropriate for supercapacitor application; nevertheless, according to the most recent information, there are quite a few different ways to synthesise MXenes [43]. The improvement of the electrode architecture is crucial to the capacity enhancement. A better volumetric capacitance of up to 1500 F cm^3^ was delivered using hydrogel Ti_3_C_2_T_x_ MXene electrodes. Excellent rate performance is demonstrated by macroporous MXene electrode designs with an open structure created using a polymethyl methacrylate (PMMA) microsphere or an MgO nanoparticle template-based technique. Additionally, the discotic lamellar liquid-crystal phase of Ti_3_C_2_T_x_ MXene can be mechanically sheared to vertically align the Ti_3_C_2_T_x_ flakes. Up to 200 micrometres, the electrochemical performance of the resultant electrode films is essentially independent of film thickness.

Additionally, the Ti_3_C_2_T_x_ films’ density could be easily raised when external pressures were applied, resulting in great volumetric performances. The ultimate structure, physiochemical characteristics, and oxidation stability of MXenes are impacted by each stage in their production. Since MXenes are topochemically produced from their parent MAX phases, the MAX phase’s quality has a significant impact on the MXenes’ structure and characteristics. The choice of the precursors has an impact on the MAX phase’s quality. The MAX phase’s raw components directly determine the characteristics of the resulting MXenes. Ti_3_AlC_2_ MAX produced from a carbon precursor based on graphite results in more conductive (4400 S cm^−1^) Ti_3_C_2_T_x_ MXene with good stability (time constant of 10.1 days). However, Ti_3_C_2_T_x_ MXene produced by MAX synthesised from the TiC-based carbon precursor is least stable (4.8 days) and relatively less conductive (3480 S cm^−1^). In a similar vein, Ti_3_C_2_T_x_ MXene produced by MAX using a lampblack carbon source is the least conductive (1020 S cm^−1^) and least stable (5.1 days). It should be noted that the synthesis of MXenes is rather difficult; however, by carefully managing all the synthesis factors, one can generate MXenes with the desired properties [44]. MXenes are typically referred to as M_n_X_n−1_ (n = 2, 3, and 4) layers, which are created by eliminating metal-ceramic MAX phase interlayer “A” atoms (where M is from transition metals, A means for elements in group IIIA or IVA, and X stands for the elements C and N). In the MAX phase, the atoms of M and X are stacked on top of one another to create a hexagonal lattice. The atoms of X occupy the centre of the M octahedral cage, which is shared by the edges of the lattice. When the atoms of A are taken out from the layer of M_n_X_n−1_, the hexagonal lattice of MX is maintained, as opposed to the cubic lattice. Consequently, the layer of M_n_X_n−1_ can be manufactured by expelling the A-atoms from the system. MXene’s thin sheets, like those of its predecessor MAX, are often laid out in a horizontal orientation. The vast majority of MXenes are well-designed mechanically and are anticipated to exhibit a high degree of longevity [45].

#### 2.1.1. Etching MAX Phase with Hydrofluoric Acid (HF)

Ti_3_AlC_2_ MAX phase was converted into multilayered Ti_3_C_2_T_x_ MXene by selectively etching the Al layers. Because of the extensive study that has been done on MXenes, etching processes have been widely utilized, particularly the method of HF acid etching, which currently is still the procedure that is utilized the most frequently. 2011 saw the publication of Naguib et al.’s proposal to manufacture the Ti_3_AlC_2_ MAX phase using HF acid etching [46]. The Ti_3_AlC_2_ MAX phase has its Al layers stripped away by the HF acid via a straightforward displacement process, which results in the generation of H_2_. Furthermore, the interaction of deionized water with the HF acid solution produces Ti_3_C_2_T_x_. (where T denotes the -O, -F, and -OH), in addition to H_2_, Ti_2_AlC, (Ti_0.5_Nb_0.5_) 2AlC, Ti_3_AlCN, Ta_4_AlC_3_, (V_0.5_Cr_0.5_) 3AlC_2_, Nb_2_AlC, and a series of the MAX complexes of Zr_3_Al_3_C_5_, Ti_3_SiC_2_, and MO_2_Ga_2_C were effectively stripped into Mxenes by using the HF acid etching [47]. The most efficient way to create MXenes materials since 2011 has been HF acid etching. This trend is expected to continue. When producing MXenes layers of superior quality, the HF acid etching process is very significant, particularly in terms of time, temperature, and number of F ions present. According to Alhabeb et al., using high concentrations of HF acid in combination with Ti_3_C_2_T_x_ results in forming a superb layered structure. This condition is difficult to accomplish with other types of acid solutions. 

The morphological and structural transition were investigated in order to further study the structural alterations of Ti_3_AlC_2_ after exfoliation. Figure 4 displays SEM images of Ti_3_AlC_2_ powders prior to and following HF treatment at various periods (2 h, 10 h, and 20 h). Figure 4a depicts the first Ti_3_AlC_2_ particle’s dense layered structure. After being submerged in HF solution for 2 h, the majority of the Ti_3_AlC_2_ layers start to separate (Figure 4b), which is quicker than the exfoliation of Ti_3_AlC_2_ powders that has been observed to occur (8 h or more) [48]. The constant elimination of Al creates more delamination as treatment time goes on, thinning the stacked sheets. This HF can be employed to exfoliate the MAX phase and create Ti_3_C_2_ MXene nanosheets, according to the study by Wang et al. as shown in the study above. 

The -O, -OH, and -F functionalities of the MXenes that were produced using the HF acid etching method were preserved, as were the distinctive surface features of the MXenes. Recent research conducted by Kim et al. [40] utilized ion-beam and electron microscopy to investigate the etching behaviour of MAX-phase Ti_3_AlC_2_ in a variety of etching chemicals at the atomic scale [50]. They analysed the difference in the structure of the Ti_3_AlC_2_ phase as an act of the etching agents and the amount of time spent etching, and their findings showed that the edge Al atoms at the mid layers of MAX-phase Ti_3_AlC_2_ are not erased, despite their interaction with the HF etchant. In addition, the grain boundary was etched while the material was treated with the HF etchant. The etching of Ti_3_AlC_2_ for three hours using the bulk etching method revealed a great number of etched zones. In addition, the SAED pattern (inset in Figure 4) shows that the d-spacing of the MAX-phase Ti_3_AlC_2_ has increased from 0.97 nm to 1.02 nm, which confirms the successful conversion to Ti_3_C_2_T_x_ MXene occurred at the etching boundary. 

#### 2.1.2. Approach Based on Modified Acid Etching

Because of the caustic and toxic nature of acid fluoride solutions, researchers are working to develop alternatives to directly use HF acid in extracting the aluminium layers from MAX phases. These alternatives are now under investigation. In the most common method, known as original-location HF acid etching, the HF acid is substituted by fluoride salts (such as LiF, NH_4_HF_2_, FeF_3_, KF, and NaF) and HCl [51]. This method was successfully done by Wang and Garnero in Issue 41, Journal of Materials Chemistry A in 2017. This technique is considered to be the most effective. In most cases, an undesirable byproduct known as AlF_33_H_2_O is produced whenever MXenes are synthesised by etching Al or Ga layers of the MAX phase with HF acid. This condition results in the formation of MXenes. It is vital to shed light on the conditions that trigger the production of this impurity so that MXenes can be synthesised without the presence of this impurity. As a result, many customised etching methods are used extensively. For example, Cockreham and colleagues [52] determined the parameters that cause the formation of the byproduct AlF_33_H_2_O while continuing the etching process using cobalt fluorides (i.e., CoF_2_/CoF_3_). These conditions resulted in the formation of AlF_33_H_2_O. On the scanning electron microscopy of the CoF_3_/MAX sample, there is no evidence of the impurity AlF_33_H_2_O. The MXenes interlayer distance that was generated by employing the modified acid-etching technique has shown significant improvement as a result of the cation’s ability to intercalate. This condition reduces the inner force exerted between the layers, and can also cause the material layers to delaminate when ultrasonication is used. Consequently, the previously mentioned laborious multi-step synthesis procedure can be shortened using this technique, making it possible to synthesise few-layer MXene in one step.

#### 2.1.3. Molten Salts Etching

MXene can also be manufactured by heating the MAX phases, such as Ti_4_AlN_3_, in a molten fluoride salt mixture (i.e., LiF, NaF and KF (29:12:59 weight ratio%)) at 550 °C while argon is used as a shielding agent. This process results in the production of Ti_4_N_3_. The process of etching can be finished in a quarter of an hour. Ti_n_N_n−1_, which has a lower stability than Ti_n_C_n−1_, can be solvated in HF or other fluoride-based acids used as the etching agent. This condition is because Ti_n_N_n−1_ is less stable than Ti_n_C_n−1_. Because of this, the etching technique that uses molten salts has the advantage of a processing time that is relatively short. In order to fulfil the requirements, additional cleaning (washing with DI H_2_O and H_2_SO4) and delamination (in TBAOH solution) must be performed. The XRD patterns of the resultant Ti_4_N_3_ show that the crystallinity of the delaminated Ti_4_N_3_ is below that of the MXene formed from the HF etching. This condition is evidenced by MXene having a higher crystallinity. The TiO_2_ phase may also be seen in the final product. The molten salts etching technique creates MXenes with finite stability in the HF or fluoride-based acids solution, compared to the HF and fluoride-based acids etching process. This condition is an advantage of the molten salts etching process. On the other hand, the following are some drawbacks associated with using this method. The resulting MXenes have small purity and crystallinity; the resulting MXenes contain many surface flaws and vacancies; the etching process uses up a large amount of heat and energy [53]. Very recently, Liu et al. reported the synthesis of MS-Ti_3_C_2_T_x_ using the intercalation of the tetrabutylammonium hydroxide (TBAOH), followed by layer separation through the use of sonication. The Cl-terminated MS-Ti_3_C_2_T_x_ that was developed was utilized as an anode in a Li-ion battery, resulting in the battery achieving a high specific capacity and an excellent rate capability. The transmission electron micrograph displays nanosheets of Ti_3_C_2_T_x_ that are transparent and have defined edges. Their lateral dimensions are approximately 600 nm.

The TBAOH treatment did not appear to have any effect on the crystallinity of the Ti_3_C_2_T_x_ material, as suggested by the fact that the SAED pattern exhibits crisp reflections that match the hexagonal crystal symmetry. The HR-TEM micrograph indicated a thickness of 2.05 nm, which is equivalent to two layers. This is in comparison to the thickness of a monolayer, which is approximately 1.03 nm. In addition to TBAOH, a number of other organic compounds, including isopropyl amine and dimethyl sulfoxide (DMSO), have been found to be efficient intercalants for the delamination of HF-MXenes. The use of TBAOH proved to be an efficient intercalant in the experiment.

#### 2.1.4. Etching without Fluoride

Although numerous etching techniques for synthesising MXenes have been validated, most synthesis approaches need HF or chemicals based on fluoride. This condition can lead to the production of -O and -F terminations on the interface of the MXene molecule. Specifically, -F terminations bring about a deterioration in the electrochemical performance of supercapacitors that are based on MXenes [54]. Therefore, fabrication methods that do not involve the use of fluoride are required in order to guarantee satisfactory electrochemical performances. Some researchers have devised an alkali-assisted hydrothermal etching process. This technique is used to manufacture of Ti_3_C_2_ MXene, and it employs a solution of NaOH as the etching agent. It is possible to employ alkali as an etchant for the Ti_3_AlC_2_ MAX phase due to the strong contact between aluminium and alkali. There is still a significant amount of difficulty involved in the process of obtaining high-purity multi-layered MXenes. The Bayer technique was applied in this scenario to etch aluminium layers without causing any harm to the Ti_3_C_2_ MXene skeleton [55]. This condition was accomplished by employing a high alkali concentration (i.e., 27.5 M) and a high temperature (i.e., 270 °C). 

Figure 5 reveals that Chen recently reported producing of a fluoride-free and chloride-containing Ti_3_C_2_T_x_ MXene by electrochemical etching. During the manufacturing process, Ti_3_C_2_T_x_ was delaminated using sonication. There was not a single harmful organic intercalant present during this step. The resulting Ti_3_C_2_T_x_ nanoflakes had a thickness of around 3.9 nm, and their dissipation in an aqueous media was extremely stable. According to theoretical expectations and actual findings, surface-attached F significantly impedes the movement of electrolyte ions and the electrochemically active regions are degraded. As a result, MXenes perform poorly when used in Li-ion batteries and supercapacitors. Therefore, it is extremely desired to manufacture MXenes utilising procedures that do not include the use of F [56,57].

#### 2.1.5. Hydrothermal Synthesis

In a high-pressure autoclave with precursor materials present, this method of producing a heterogeneous reaction includes heating aqueous solutions to a temperature higher than the point at which water reaches its boiling point. It is possible to manipulate the size, shape, morphology, and characteristics of QDs by taking advantage of the synergistic effect produced by increasing the temperature, the pressure, and the pH of the solution [58]. In addition, the generation of MXene is heavily dependent on the reaction’s temperature, the solution’s pH, and the amount of time that passes. Because changes in pH and temperature can affect the length of time it takes for a reaction to take place, the temperature range for the MXene synthesis must be kept constant at between 100 and 180 °C (C) [59]. In addition, the material’s size, characteristics, and thickness can be changed by modifying the circumstances under which the hydrothermal reaction occurs. 

Xue used a hydrothermal method to manufacture water-soluble Ti_3_C_2_ MXene and noticed that the characteristics, size and thickness of MXene can be customised by adjusting the temperature of the hydrothermal reaction to 79 °C. This condition allowed Xue [60] to create water-soluble Ti_3_C_2_ MXene. The MXenes produced at temperatures of 100, 120, and 150 °C created particles with average diameters of 2.9, 3.7 and 6.2 nm, respectively, and produced particles with average thicknesses of 0.99, 0.91 and 0.89 nm correspondingly, indicating that monolayers make up the majority of the particles. The Ti_3_C_2_ QDs display the -NH surface functions during the reaction, and at a lower temperature (i.e., 100 °C), a new MXene structure is generated, in which the d-spacing value may be validated. On the other hand, the MXENE generated at a temperature of 120 °C displays a fusion structure with CTi located at the core and TiO_2_ on the surface. An amorphous structure of MXene was produced, despite the fact that the majority of the Ti atoms had been removed from the compound by etching at a higher temperature (namely, 150 °C). 

According to Xiao [61], the MXene structure cannot be formed at temperatures lower than 100 °C. They produced MXene (Ti_3_C_2_T_x_) by adjusting the reaction temperature from 60 to 80 and 100 °C. At 60 °C, many nanoribbons, together with a few Ti_3_C_2_T_x_ nanoflakes, were created. MXene was not the product of this reaction. The particles shrank in size as the temperature was raised to 80 °C, and nanodots of varying sizes were produced as a result of this process. At a temperature of 100 °C, ultrafine nanodots are formed and then spread out equally across the nanosheets. By utilising a variety of hydrothermal conditions, other MXenes have been successfully synthesised [62,63]. MXene has also been created with doped heteroatoms utilizing the elements’ subsequent precursors. This condition was accomplished by modifying the conventional hydrothermal conditions in order to manufacture it. Xu and his colleagues manufactured heteroatom-co-doped Nb_2_C MXene (S, NMXene) by employing the hydrothermal process at 160 °C. 

The precursor material for this material was Nb2C nanosheets, and the source of sulphur and nitrogen was L-cysteine. MXene has particles ranging in size from 2.6 to 4.7 nanometers in diameter. The lateral size of the synthesised S, N-MXene is 3.56 nm, which is noticeably smaller than the lateral sizes of the MXene (which is 2.4 nm) and the N-MXene (i.e., 2.66 nm). It was determined from the fact that the S, N-MXene had an average thickness of 1.74 nm that a monolayer had been formed [64]. Recently, Peng successfully created 2D MXene (also known as h-Ti_3_C_2_) by employing the hydrothermal technique and etchants of low toxicity (i.e., NaBF_4_, HCl). A layered structural morphology could be seen in the transmission electron microscopy of the resulting h-Ti_3_C_2_ MXene. Figure 6 demonstrates that the 2D h-Ti_3_C_2_ nanoflakes have very low thickness and are completely see-through. Figure 6 shows a selected area HR-TEM micrograph of h-Ti_3_C_2_ nanoflakes, which reveals a d-spacing of 0.264 nm and 0.155 nm. These values correspond to the (0110) and (0210) planes of the Ti_3_C_2_, respectively. As may be observed from the inset of Figure 6, the FFT pattern exhibited hexagonal symmetry. It is essential to consider that the hydrothermal etching process is effective in preventing the consumption of HF acid and a time-saving approach to the production of Ti_3_C_2_ nanoflakes.

### 2.2. Preparation of MoS_2_

The synthesis of MoS_2_ nanostructures can be accomplished using a variety of different methods. In most cases, either a “top-down” or a “bottom-up” technique can commonly be used in the preparation of MoS_2_ nanoparticles. In summation, both approaches are displayed in the Figure 7. In this review, it will highlight the top most acceptable methods that are related to the application of supercapacitors.

#### 2.2.1. Mechanical Exfoliation

This is one of another type of technique, which is also known as “micromechanical cleavage”, was the first approach utilized to generate two-dimensional material from bulk layered materials by mechanical fragmentation. This was accomplished by the process of mechanical fragmentation. This method was used to prepare graphene [66]. In addition, the process of mechanical exfoliation, commonly known as the Scotch-tape method, can be defined as the “detaching” or “peeling” of bulk crystals by employing adhesive tape or by bulk crystals rubbing against a solid surface. Both of these processes are examples of mechanical exfoliation [67]. This approach is also known as the mechanical method. Huang et al. in 2005 found that, when the substrate was cleaned before the tape was adhered to, the results were much improved. Considering gold has a good tolerance for chalcogens, which effectively overcomes the van der Waals force that exists between the top layer and the remnant of the crystal, mechanical exfoliation can be improved by often utilising a film of gold as an intermediary substrate [68]. The most efficient method is mechanical exfoliation, which does not require expensive or specialised equipment to create the most crystalline, cleanest, and atomically thin nanosheets of layered materials because mechanical exfoliation does not require abrasive particles to be removed from the surface of the material [69]. Nevertheless, because this technique cannot be applied for processing on a large scale, the only application for which it can be utilized is for the preparation of samples for research.

Recent research has shown that a process known as nanomechanical exfoliation, an extension of mechanical exfoliation, may produce high-quality MoS_2_ nanosheets with a distinct layer. In the process of nanomechanical exfoliation, an extremely sensitive tungsten probe with a tip width of approximately 10 nm is utilized. This probe is then used to peel nanosheets off a thick fake of MoS_2_ that has been deposited on the substrate with an edge-on alignment. This probe has a diameter of about 10 nm. The tungsten probe is powered by piezoelectric actuators, and the entire operation can be observed in real time using a high-resolution transmission electron microscope [70]. Miyake et al. processed silicon at the nanometer scale on a device with a radius of less than 50 nm using an atomic force microscope [71].

#### 2.2.2. Electrochemical Exfoliation

The electrochemical exfoliation process is an intriguing and also possibly useful procedure often carried out in mild settings; it is also straightforward, easy, repeatable and may be suitable for manufacturing on a big scale [72]. Exfoliation of bulk multilayer MoS_2_ was accomplished by an electrochemical process known as lithium intercalation (cathodic exfoliation of MoS_2_). Lithium and MoS_2_ metal foil were employed to create the cathodic and anodic poles, respectively. When discharging at a steady current, lithium ions will intercalate within the bulk stratified MoS_2_, reducing the strength of the van der Waals force between the layers. The intercalated compounds are rinsed with acetone and then ultrasonically treated in either water or C_2_H_5_OH to exfoliate and remove the two-dimensional nanosheets [73]. It is difficult to remove the residual lithium doping effect using this technique, despite the fact that it can exfoliate multi-layered materials into monolayers (for example, single-layer MoS_2_), which results in the MoS_2_ nanosheets losing their semiconducting properties due to the lithium still being present in the MoS_2_ nanosheets [74]. Electrochemical anodic exfoliation of bulk MoS_2_ was employed by Liu et al. in order to create high-quality thin nanosheets of MoS_2_ [75]. A bulk layered MoS_2_ crystal served as the anode, while a Pt thread served as the counter electrode, and the electrolyte was an aqueous solution of 0.5 M Na_2_SO_4_. Excellent consistency and intrinsic structure can be found in the exfoliated monolayer and few-layer MoS_2_ nanosheets with lateral dimensions of up to 50 m [76].

#### 2.2.3. Hydrothermal Synthesis

This is a type of method which conventionally applied a wet-chemical synthesis approach that produces nanomaterial with controllable size, high yield and homogeneous layer thickness. It operates at a high temperature of vapour pressure in an autoclave while under high pressure, and it does so at a high temperature. The only distinction between hydrothermal and solvothermal synthesis is that hydrothermal synthesis requires an aqueous precursor [77]. Ionic liquids may be a helping hand in the preliminary stages of the hydro/solvothermal synthesis process. Micro-spheres of MoS_2_ with a diameter of 2.1 mm by a hydrothermal procedure that involved the use of 1-butyl-3-methylimidazolium tetrafluoroborate. Du et al. used a solvothermal process that involved the help of an ionic liquid in order to create MoS_2_ nanospheres [78]. They utilized 1-ethyl-3-methylimidazolium bromide as a template in solvents that contained a mixture of dimethyl formamide and water. The morphology and size of the MoS_2_ formed are both significantly impacted by the ratio of dimethyl formamide to water that is used. The production of few-layered MoS_2_ nanosheets was accomplished using a surfactant-assisted hydrothermal method [79]. According to the findings, the artificially produced MoS_2_ nanosheets’ morphology was reminiscent of petals.

#### 2.2.4. Microwave Synthesis Method

The synthesis method for producing MoS_2_ is uncomplicated, risk-free, and productive concerning time and energy [80]. The approach of a microwave-assisted processing path was used to manufacture nanotubes as well as fullerene-like MoS_2_ nanoparticles. The amorphous powders of MoS_2_ were calcined for two hours at a temperature of 600 °C in the presence of argon. After 200 s of microwave irradiation, the synthesised MoS_2_ has a structure similar to that of fullerene, with orientated at random, strongly doubled-up layers of MoS_2_ ordered in a short range. Irradiation for a more extended period (600 s) produced nanostructures with a morphology similar to that of nanotubes and fullerene [81]. The formation of MoS_2_/poly (ethylene glycol) nanoflowers was accomplished by a hydrothermal method that was helped by microwaves. Irradiation with microwaves at a temperature of 220 °C was applied to the MoS_2_ precursors for ten minutes. The nanoflowers form of molybdenum disulfide/poly (ethylene glycol) is composed of many layers of molybdenum disulfide nanosheets [82]. Researchers used a microwave-initiated approach that was safe, simple, scalable, rapid, and efficient. This method did not entail the use of inert gas in any way. They reported a direct development of MoS_2_ on graphene substrate [83]. The dried mixture of (NH_4_)_2_MoS_4_-graphene-CS_2_ was irradiated in a microwave oven for 60 s in order to make it suitable for home usage (2.45 GHz, 1250 W). Graphene was utilized as a substrate, absorbed microwave radiation and converted it to heat. This heat reduced (NH_4_)_2_MoS_4_ to MoO_2_, which led to the transformation of graphene into MoS_2_ that was scattered on the graphene substrate. The standard hydrothermal and microwave procedures produce MoS_2_ nanosheets, and their electrocatalytic performance for hydrogen evolution is compared. Despite the fact that the hydrothermal synthesis procedure takes 24 h, the microwave-assisted synthesis of MoS_2_ only takes 30 min. The MoS_2_ synthesised by either method has equivalent crystal structural features. Both methods produce thin and joined nanosheets, but the MoS_2_ nanosheets made in a microwave have a smoother edge and a less crumpled shape than those produced by the other method. Both MoS_2_ nanosheets exhibited electrocatalytic performance that was almost identical [84]. There are few techniques for synthesising the material such as using aerogel, liquid assisted sonication, liquid exfoliation and ultrasonic cavitation, exfoliation of and ultrasound sonication in supercritical CO_2_, supercritical CO_2_ exfoliation and sonication, atomic layer deposition (Ald) and thermal evaporation, chemical vapour deposition (Cvd) and liquid precursor, Ald and thermal evaporation, self-propagation of Mo powder and elementary sulfur, Cvd with sulfur as a precursor (sulfidation), sulfidation and intercalation and exfoliation [85].

### 2.3. Preparation of Ti_3_C_2_ MXene/MoS_2_

Hydrothermal synthesis constitutes the majority of the effective methods for the manufacture of hybrid MXene/MoS_2_, despite the fact that there have recently been a lot of academics saying there are other ways to synthesise it. In this review, we will provide an explanation in terms of the methodology that is appropriate for the objectives of application. Li et al. in 2021 stated that, because of the synergistic impact, the unique 2D/2D heterostructures can fully combine their separate 2D features and demonstrate better performance. However, research on how to properly incorporate other foreign 2D materials on the 2D MXene substrate, as well as detailed synergistic effects, is currently lacking. Impressively, the unique off-axis electron holography is first employed to visually characterise the charge density distribution at the 2D interface, which establishes an effective relationship between the charge density distribution at the 2D/2D heterostructures. The 2D/2D MXene-MoS_2_ composites are successfully made using a simple hydrothermal process, as shown in Figure 8.

#### 2.3.1. Hydrothermal Synthesis Method

According to Chandran in Journal of Energy Storage in 2020, the MoS_2_/MXenes heterostructures were made in the following way: an adequate quantity of ammonium heptamolybdate was solvated in a mixture of ammonium hydroxide (60 mL) and water (5 mL), H_2_S gas was purged at 25 °C for 60 min, and then gradually the temperature was increased up to 60 °C, at which point the mixture turned a dark red colour [86]. After the as-prepared MXene had been disseminated in the dark red mixture for incipient wetness impregnation, the slurry was heated at 300 °C under a nitrogen atmosphere to obtain MoS_2_/MXene [87].

Prepare to make the 1T-MoS_2_/Ti_3_C_2_ MXene heterostructure from Wang et al. in 2020. After dissolving thiourea and (NH_4_)_6_Mo_7_O_2_44H_2_O in 21.8 mL of deionized water (this step is necessary for the synthesis of 2H-MoS_2_), 0.09 g of C_6_H_8_O_7_H_2_O and 0.13 g of Ti_3_C_2_ MXene were added to the solution, followed by steady stirring [88]. After that, the homogenous solution was placed into an autoclave lined with Teflon and stainless steel (28 mL in capacity). The subsequent experimental technique is compatible with the synthesis of 1T-MoS_2_, completed after that. The mass of the 1T-MoS_2_/Ti_3_C_2_ MXene powder once produced is approximately 0.63 g. As a result, the mass percentage of Ti_3_C_2_ MXene found in 1T-MoS_2_/Ti_3_C_2_ MXene is around 20.6%, and the mass ratio of 1T-MoS_2_ to Ti_3_C_2_ MXene is approximately 79.4%:20.6%. The GCD curves of the electrodes are shown in Figure 9 at a current density of 2 A g^−1^. It can be seen that the 1T-MoS_2_/Ti_3_C_2_ MXene electrodes can deliver a high capacitance of 386.7 F g^−1^ at 1 A g^−1^, indicating that the inclusion of 1T-MoS_2_ can significantly increase the specific capacitance of Ti_3_C_2_ MXene.

Electrochemical impedance spectroscopy (EIS), galvanostatic charge–discharge (GCD) and cyclic voltammetry (CV) measurements were used to examine the electrochemical performance of all samples. On the 2H-MoS_2_, Ti_3_C_2_ MXene, and 2H-MoS_2_/Ti_3_C_2_ MXene electrodes, CV and GCD curves were performed as shown in Figure 10. Due to the low capacitance of 2H-MoS_2_ (19 F g^−1^) and Ti_3_C_2_ MXene (26 F g^−1^), as previously reported, MXene exhibits comparatively poor electrochemical performance of 36 F g^−1^. The evident increase in specific capacitance will result from the greater interlayer spacing, increased hydrophilicity, and higher conductivity of 1T-MoS_2_ compared to 2H-MoS_2_. The 1T-MoS_2_/Ti_3_C_2_ MXene electrodes have an extremely high specific capacitance that is ten times more than that of the 2H-MoS_2_/Ti_3_C_2_ MXene electrode. The CV area of the 1T-MoS_2_/Ti_3_C_2_ MXene electrodes is the greatest at a scan rate of 20 mV s^−1^ and other chosen scan rates as shown in Figure 10c, indicating the highest specific capacitance. The GCD curves of the electrodes are shown in Figure 10d at a current density of 2 A g^−1^. It can be seen that the electrodes, which are made of 1T-MoS_2_/Ti_3_C_2_ MXene, can deliver a high capacitance of 386.7 F g^−1^ at 1 A g^−1^, indicating that the addition of 1T-MoS_2_ can significantly increase the specific capacitance of Ti_3_C_2_ MXene.

One of the most crucial factors in determining how well a supercapacitor performs is cycle stability. A cycle test with a high current density of 50 A g^−1^ was conducted to look into the stability of 1T-MoS_2_/Ti_3_C_2_ MXene in electrochemical testing. In addition, 96.4% of the initial capacitance was maintained after 10,000 cycles, as shown in Figure 11, demonstrating that the 3D network nanostructure of 1T-MoS_2_/Ti_3_C_2_ MXene is extremely stable for quick energy storage. After 20,000 long-term cycles, the cycling stability is marginally improved to 96.8%. The steady activation of the surface is what caused the marginal capacity increase from 10,000 cycles to 20,000 cycles, which is consistent with earlier findings.

Yao et al. in 2020 have also synthesised a very excellent MXene/MoS_2_ with this technique. First, solution A is created by dissolving 1.1 g of ammonium molybdate ((NH_4_)_6_Mo_7_O_24_·4H_2_O) and 2.2 g of thiourea ((NH_2_)_2_CS) in deionized water for 60 min while vigorously stirring. Next, a quantity of Ti_3_C_2_ nanosheets is added to 20 mL of deionized (DI) water, which is then stirred for 30 min before undergoing an additional 40 min of ultrasonication. This mixture is referred to as solution B. Then, A and B are combined drop by drop while being ultrasonicated for 30 min. The combined solution is put into a 100 mL Teflon-lined autoclave and kept there for 7 h at 180 °C. The black catalysts are cleaned with DI water three times to remove the dispersion agent and then dried at 70 °C for 10 h in a vacuum oven. The mass ratio of Ti_3_C_2_ MXene to MoS_2_ is set to 0, 0.1%, 0.3%, 0.5%, 1.0%, and 2.0wt%, respectively, by adding the Ti_3_C_2_ solution. The prepared samples have a unique label, such as TM0, TM0.1, TM0.3, TM0.5, TM1, and TM2.

#### 2.3.2. Ultrasonic Treated Method

Another recent method from Lui in 2022 involved 7.5 mL MXene dispersion (2 mg/mL) and 15 mL MoS_2_ dispersion (20 mg/mL) being dissolved in beakers. After adding 22.5 mL of deionized water, the mixed solution was diluted to a concentration of 4 mg/mL. After mixing the solution, it was subjected to ultrasonic treatment for 15 min. The elevated temperature reducing self-assembly approach was utilized in synthesizing MXene/MoS_2_ microspheres by ultrasonic atomizer. The nozzle had an ultrasonic power of 1.5 W when it was being utilized. The rate of flow in the combined solution was 0.5 mm per minute. The water vapour was turned into vapour by injecting liquid paraffin into the drop of liquid produced through ultrasonic atomization at a temperature of 150 °C. The method of centrifugation was utilized in order to collect the MXene/MoS_2_ microspheres. After being cleaned with n-hexane and 100% ethanol, they were ultimately positioned inside a fume hood to be dried. MXene/MoS_2_ microspheres with mass ratios of MXene and MoS_2_ of 3:1, 5:1, and 7:1 was gained by adjusting the ratio of raw materials. These microspheres were given the names MXene/MoS_2_-3, MXene/MoS_2_-5, and MXene/MoS_2_-7. The concentration of the mixed solution was kept the same throughout the process [89].

## 3. Material Characterization

Material characterization is the foundation for knowing the composition of an energy storage device material and its potential to cause a good or bad effect when the device is used. A single-layer with high conductivity and a multilayered with low conductivity are the two morphologies of MXene that are produced from their parent MAX due to the differing etching method. The exploration of multilayered MXene, which has a unique accordion-like structure, offers natural structural benefits and the multiple dispersion of electromagnetic waves, is the focus of the majority of the effort. Many studies have recently been conducted to widen the multilayered MXene’s layer spacing and boost its conductivity.

### 3.1. Morphology and Nanostructure

Basic characterization protocols are required to ensure that results may be reproduced accurately, despite the fact that various methodologies and protocols have been described in numerous papers written by members of the 2D materials community. In addition, for the majority of typical 2D materials, field-emission scanning electron microscopy (FESEM) and transmission electron microscopy are two techniques that can be utilized in order to investigate the morphology and nanostructure of the hybrids (TEM). The MXene-MoS_2_ interface and the unique lattice fringes can be demonstrated by gaining a deeper understanding of the nanomaterial [90].

Raman has previously been standardised (from monolayer to an increasing number of layers). You may forecast the quantity of 2D layers and their crystalline clarity, flaws, etc. using optical microscopy and Raman. At the conclusion, we can utilise AFM/TEM to confirm. The acquisition of selected area electron diffraction is currently the sole way to discern the number of layers in a transmission electron microscope (TEM) (SAED). This does not result in a direct thickness measurement that can be used to determine how many 2D monolayers are present in an isolated stack. The number of monolayers multiplied by the stated thickness value for a single layer can therefore be used to estimate the thickness. This method can be challenging because it does not explicitly disclose any chemical information, making it difficult for the user to distinguish between various 2D materials and impurities. It is only possible to chemically characterise 2D materials while imaging them when electron microscopy and energy dispersive X-ray spectroscopy are used together (EDS). The layer number is determined using a variety of techniques, including Raman spectroscopy, photoluminescence (PL), atomic force microscopy (AFM), and optical contrast. Equipment and systems specifically designed for Raman spectrometry, AFM, and PL setups are required. Because of this, standard layer identification methods are slow and expensive, especially for multilayered samples. The optical contrast method, on the other hand, is incredibly effective and inexpensive because it just needs a simple optical microscope imaging setup. Layer numbers in two-dimensional materials have been identified using optical microscope pictures in numerous earlier research. Even though MXene/layer MoS2’s count and thickness are significant indicators, only a small number of researchers who focus on this particular topic, which has the supercapacitor as its application, have talked about it. We come to the conclusion that other indications deserve more attention.

Based on Chien et.al. in 2017, Figure 12 obtained using scanning electron microscopy (SEM) of MoS_2_/MO_2_TiC_2_T_x_-500 and MoS_2_/MO_2_TiC_2_T_x_-700 are presented in Figure 12a,b, respectively. MoS_2_/MO_2_TiC_2_T_x_ heterostructures are disordered and have an open architecture formed of 2D sheets, in contrast to pure MO_2_TiC_2_T_x_, created by filtering of a stable colloidal solution and is composed of neatly aligned stacked 2D sheets. Figure 12c displays the pictures that transmission electron microscopy (TEM) produced when applied to MoS_2_/MO_2_TiC_2_T_x_-500. It was found that the lattice spacing was less than 14.0 angstroms, which is consistent with the position of the (002) peak in the XRD pattern of the MO_2_TiC_2_T_x_. In addition, a newly created layered compound was discovered on MO_2_TiC_2_T_x_. The interlayer spacing of this compound was determined to be approximately 6.9 angstroms, which is comparable to that of bulk MoS_2_ (6.15 angstroms). This value is consistent with the d-spacing that we determined using the (002) peak of MoS_2_ in the XRD patterns, from whence it was derived. Figure 12c demonstrates that two layers of MoS_2_ make intimate contact with layers of MO_2_TiC_2_T_x_, resulting in the formation of MoS_2_-on-MXene heterostructures. It is clear from the transmission electron micrograph of MoS_2_/MO_2_TiC_2_T_x_-700 that there are a greater number of MoS_2_ layers, which suggests that a higher temperature of 700 °C makes the synthesis of MoS_2_ easier.

Guo et al. in their research stated that layers of titanium, carbon and aluminium are securely packed together without any gaps or splits to form the unetched MXenes material. The outcomes of the EDS examination can also be used to determine the composition of the titanium, carbon, and aluminium atoms [91]. The morphology of Ti_3_C_2_ and Ti_3_C_2_-MoS_2_ composites was studied by SEM, as illustrated in Figure 13A,B. On the contrary, it was visually apparent that many lamellar materials were stacked on top of one another layer by layer, demonstrating that Ti_3_AlC_2_ had been successfully etched by removing the aluminium atom layer after being subjected to HF treatment while simultaneously creating two-dimensional multilayered Ti_3_C_2_. From Figure 13C, it was clear that Ti_3_C_2_-MoS_2_ made by the hydrothermal process had a thick layer of flower-like materials on the surface of the layered Ti_3_C_2_. Figure 13D contains more precise structural details concerning Ti_3_C_2_-MoS_2_ composites. MoS_2_ sheets placed on Ti_3_C_2_ should have the shape of cascading petals. We were able to determine that the surface area of the modified Ti_3_C_2_-MoS_2_ had been enlarged with a straightforward chemical alteration by comparing the SEM images of Ti_3_C_2_ and Ti_3_C_2_-MoS_2_. Additionally, as can be observed from the particular surface measurement used in the study, Ti_3_C_2_-MoS_2_ has a specific surface area that is 60.49% larger than Ti_3_C_2_. For Ti_3_C_2_-MoS_2_ composites’ subsequent adsorption uses, the creation of composites should be of utmost importance.

### 3.2. Specific Surface Area

The Brunauer–Emmett–Teller (BET) technique, which involves the adsorption and desorption of nitrogen gas, can be utilized to calculate the specific surface area of a dried MXene/MoS_2_ hybrid. Chandram and Thomas in 2020 stated, because MoS_2_ was incorporated into MXene, the BET analysis showed the proportional differences in the surface area and pore volume of MoS_2_/MXene.

From the recent study done by Lui et al. in 2022, three samples of MXene/MoS_2_ were tested by this type of testing. N_2_ adsorption equipment was used to survey the pore characteristics of MXene/MoS_2_. The adsorption–desorption isotherms were type IV, as shown by Figure 14. It demonstrated that the samples contained both mesopores and macropores. The aforementioned holes were made by stacking layers of two-dimensional material. The slit channel reflected by the H_4_ hysteresis loop could support this. Table 2 showed the MXene/MoS_2_ pore performance metrics.

Folded microspheres’ pore size and volume were first increased and then reduced as the MXene filler ratio was increased, whereas the BET surface area first decreased and then increased.

### 3.3. Binding Nature

Patterns of X-ray diffraction (XRD) of Mxene films will be obtained by employing CuK radiation with a powder diffractometer (PANANALYTICAL). In order to gather the XRD patterns of Mxene powders/films and MAX powders, a powder diffractometer equipped with Cu K radiation will be employed, and the step size will be 0.03°, and the dwelling time will be 0.5 s. Analyses will be performed on hybrid samples before and after electrochemical testing for comparison. In order to characterize the crystallographic structures and the binding nature (chemical state) of the materials, an X-ray diffractometer (XRD) and an X-ray photoelectron spectrometer (XPS) will be utilized. Binding energies will be referenced to the C 1s peak of the (C-C, C-H) bond, which will be set at 284.8 eV. Chemical compositions and the oxidation state of the samples will be further investigated using high resolution XPS with monochromated Al K radiation (hv = 1486.6 eV). The peak fitting will be done with the assistance of CasaXPS, a software that is available for purchase.

The MoS_2_/MXene XPS spectra displayed in Figure 15 was completed by Chandran et al. The survey spectra show the peaks that signify the presence of the MoS_2_/MXene components Ti 2p, C 1s, O 1s, Mo 3d, and S 2p High-resolution XPS spectra of each element, providing confirmation of the electronic states that are accessible on the element’s surface. Figure 15b shows the Ti 2p spectrum, which has four peaks that fit together and are located at binding energies of 454.5, 458.5, 460.1, and 464 eV. These peaks correspond to Ti-C (Ti 2p1/2), Ti-C (Ti 2p3/2), Ti-O (Ti 2p1/2), and Ti-O (Ti 2p3/2) peak, respectively, and may be involved in the construction of TiC, TiCO, and TiO_2_ bonds. The total peaks of four at 284.2, 285.80, 289.35, and 285.06 eV were ascribed to C-Ti, C=C, CC, C-O, and/or the C-S, respectively, in the C 1s region of the XPS spectra of MoS_2_/MXene (Figure 15c). Peaks that suit the MoS_2_/MXene O 1s profile at 530.8, 531.8 and 532.5 eV, respectively, correspond to Ti-O_x_ and/or the Mo-O_x_, Ti-(OH)_x_, and CO (Figure 15d), revealing the synthesis of MoS_2_/MXene. High solution XPS spectra of Mo 3d also served to confirm the in situ production of MoS_2_ (Figure 15e). The two most prominent peaks of the Mo 3d may be found at 229.5 and 232.6 eV, which are the corresponding Mo^4+^ 3d5/2 and Mo^4+^ 3d3/2 of MoS_2_, respectively. Additionally, a peak at 226.7 eV belongs to MoS_2_’s S 2s, and a peak at 235.8 eV is associated with Mo^6+^, which is the result of a minor oxidation of Mo edges in MoS_2_ from the Mo^4+^ state to the Mo^6+^ state [23]. The development of MoS_2_ on the MXene layers is further supported by Figure 15f, which shows S 2p spectra that are contoured for two peaks at 162.31 and 161.5 eV, which are attributed to S 2p1/2 and S 2p3/2, respectively.

Wang and Li et al. in 2020 concluded that the d-spacing can be derived from the XRD result through the Bragg formula. According to the findings, the d-spacing for the MoS_2_ on Ti_3_C_2_ MXene that was prepared with a magnetic field of 9T is smaller than 6.3 angstroms and is 9.4 angstroms for the sample that was prepared with no magnetic field. The XRD peak positioned at 2 of 25.3° can be identified as the (101) peak of TiO_2_, typically seen in the hydrothermal processing of Ti_3_C_2_ MXene. Additionally, the primary peak (002) of 1T-MoS_2_ is at 9.3° in Figure 16d. This condition shows that the increased interlayer spacing leads to the facilitation of the intercalation. It is interesting to note that the diffraction peaks at 7.1° and 9.3° in XRD patterns can be seen clearly in 1T-MoS_2_/Ti_3_C_2_ MXene. These peaks correspond to the (002) peaks of 1T-MoS_2_ and Ti_3_C_2_ MXene, respectively. The position of the (002) peak of Ti_3_C_2_ MXene in 1T-MoS_2_/Ti_3_C_2_ MXene is placed at 7.1°, which is consistent with that in 2H-MoS_2_/Ti_3_C_2_ MXene. This condition can be seen in Figure 16e. It is possible to identify the (004) peak of MoS_2_ as the XRD peak placed at two angles of 28.6°.

### 3.4. Quality and Purity

Raman measurements will be carried out on the samples utilising a Raman spectrometer (UniRAM-3500) with 532 nm laser excitation. These measurements will be utilized to observe the quality and purity of the MXene/MoS_2_ compound. Raman analysis will be employed as a method of evaluation to evaluate the yield (amount) of the sample, in addition to conducting a wide-ranging inquiry on their diameter and electronic properties. In the Raman spectrum of MoS_2_ nanostructures, three distinct characteristic band positions are anticipated to be seen. These band positions are designated E1g, E2g, and A1g, located at 296.86, 346.87, and 390 cm^−1^ correspondingly. The presence of a S atom in the basal plane is the cause of the Raman band having vigorous intensity and appearing at 296.86 cm^−1^; this band has E1g symmetry. The intralayer vibrational mode of Mo and S atoms in the basal plane is responsible for the formation of the band that appears at 347.86 cm^−1^ (E2g symmetry). The intralayer mode that involves the mobility of S atoms is the cause of the appearance of the A1g mode at a frequency of 390 cm^−1^. Together with MXene, the MoS_2_ nanostructures will significantly increase the number of exposed electrochemically active sites, which will, in turn, significantly improve the efficacy of ion transport during reversible electrochemical reactions. The energy density of a supercapacitor can be increased by regulating the porosity of the active material. The porosity of the active substance should be either larger than the desolvated ions or equivalent to or smaller than the hydrodynamic size of the active ion. Because of the porous nature of the active material, it will be possible to slow the rate of discharge, which will result in an increase in the supercapacitor’s energy density.

As reviewed in the same paper, performing Raman spectra on 2H-MoS_2_/Ti_3_C_2_ MXene and 1T-MoS_2_/Ti_3_C_2_ MXene was completed by Wang et al. so that the exact crystal structures of both of these compounds could be better understood. As can be seen in Figure 17c,f, for the 2H-MoS_2_/Ti_3_C_2_ MXene and 2H-MoS_2_, two characteristic Raman peaks of E2g 1 and A1g at the wavelengths of 377 and 403 cm^−1^ are clearly detected. This observation lends credence to the idea that the 2H phase is present. In contrast, it is possible to make out three typical Raman peaks at 147 cm^−1^ (J1), 235 cm^−1^ (J2), and 335 cm^−1^ (J3), which validates the presence of the 1T phase of MoS_2_ in the aforementioned heterostructure by using the magneto-hydrothermal approach. In addition, the presence of the E1g Raman peaks at 280 cm^−1^ verifies that the octahedral coordination of Mo is present in the 1T-MoS_2_/Ti_3_C_2_ MXene heterostructure. Both of the surfaces of the Ti_3_C_2_ MXene are covered by a nanosheet composed of 1T-MoS_2_ or 2H-MoS_2_, as can be seen in the SEM photos that follow.

### 3.5. Electrochemical Properties of MXene/MoS_2_

Chandran et al. stated in 2020 that, by adopting a symmetric two-electrode system and cyclic voltammetry, the electrochemical performance of MXene and MoS_2_/MXene was examined. The cyclic voltammogram of MoS_2_/MXene and MXene at various scan speeds of 20, 50, and 100 mVs^−1^ in the potential window of –1.5 to 1.5 V are shown in Figure 18a. It is clear that both MXene and MoS_2_/MXene exhibit a quasi-rectangular CV response, demonstrating the materials’ strong capacitive nature. The MoS_2_/MXene in aqueous H_2_SO_4_ electrolyte exhibits oxidation/reduction in the potential window of –1.5 to 1.5 V, and the cell voltage is around 3 V. In general, cell voltage for aqueous electrolyte solutions is about 1 V, and, for organic electrolytes, it is around 3–3.5 V. Due to the synergetic effect of MXene and MoS_2_, the potential window is increased, making these MoS_2_/MXene electrodes suited for high power applications. No significant oxidation or reduction peaks in the CV responses of MXene and MoS_2_/MXene were seen as the scan rate was increased, demonstrating that these electrodes are charged and discharged at a pseudo-constant rate during the whole cycle. It appears that MXene and MoS_2_/MXene are electrochemically stable at high potentiality since the area of the CV curve for these materials increases with increased scan rate without deforming its overall rectangular shape. Because of the resistance that MoS_2_ provides, the effective contact between the ions and the electrode is significantly decreased at faster scan rates and is hence stable at high potential. Because MXene and MoS_2_ work together synergistically to increase specific capacitance, Figure 18b demonstrates that the area of the CV curve for MoS_2_/MXene electrodes is much higher than that of MXene electrodes. The stacks of MXene are filled with MoS_2_ during the confinement of MoS_2_, and the synergistic impact of MoS_2_/MXene adds additional active sites for reversible actions, reduces internal resistance, and improves ion transfer and pseudocapacitance behaviour.

Chen et al. in 2019 successfully synthesise direct laser etching free-standing MXene-MoS_2_ film for a highly flexible micro-supercapacitor. For portable and miniature electronics, micro-supercapacitors (MSCs) with high electrochemical performance and outstanding flexibility are crucial. In this paper, the free-standing MXene-molybdenum disulfide (MoS_2_) film for MSC is directly etched using a laser etching technique that has the advantages of being a straightforward procedure, low cost, and having high machining accuracy. The electrochemical performance of MXene is significantly enhanced by the addition of MoS_2_, as seen by its higher specific capacitance (which is around 60% more than pure MXene). Finally, the highest specific capacitance of this MSC is 173.6 F cm^−3^ (1 mV s^−1^) based on the combined volume of the positive and negative electrodes, the maximum power density is 0.97 W cm^3^, and the maximum energy density is 15.5 mWh cm^−3^. The MSC also exhibits exceptional cycle stability and flexibility; for instance, its capacitances still hold roughly 98% and 89% of their initial capacitance after 6000 charge–discharge cycles and 150° of bending, respectively. A novel concept for upcoming high-performance micro energy storage devices is provided by the laser-etched MSC based on MXene-MoS_2_.

For the conclusions of their research, the free-standing finger-like MXene-MoS_2_ electrodes were prepared using a laser etching technology, which has the advantages of a straightforward process, low cost, and high machining accuracy. An MSC was produced when the finger-like MXene-MoS_2_ electrodes were transferred to PDMS film, which was then coated with gel electrolyte. The synthesised MSC finally displays strong electrochemical performance because adding MoS_2_ to MXene significantly increased the electrochemical performance, such as a greater specific capacitance (approximately 60% higher than pure MXene). For instance, this MSC has a maximum specific capacitance of 173.6 F cm^3^ (1 mV s^−1^) based on the combined volume of the positive and negative electrodes, a maximum energy density of 15.5 mWh cm^3^ and a maximum power density of 0.97 W cm^3^, and a capacitance that retains 98% of its initial capacitance after 6000 charge–discharge cycles. The MSC also has great flexibility; for instance, when bent at an angle of up to 150°, a high capacitance retention of 89% may be attained. In conclusion, MSC may be produced using an easy-to-implement manufacturing approach, and the finished MSC based on MXene-MoS_2_ demonstrates great flexibility and high electrochemical performance.

Using GCD measurements, the specific capacitance of MXene and MoS_2_/MXene was examined. Figure 19 depict the GCD curves for MoS_2_/MXene and MXene at various current densities of 0.4, 1.0, 1.6, 2.0, and 4.0 A g^−1^. The nearly symmetrical triangle shown on the charging and discharging GCD curves of MoS_2_/MXene and MXene, along with the linear voltage/time profiles, show the device’s excellent capacitive performance and quick, reversible faradaic reactions. The same pattern can also be seen in the GCD curve of MoS_2_/MXene at high current densities (10, 12, and 20 Ag^−1^). The charging and discharging processes result in a minimal internal resistance (IR) decrease and symmetric charge and discharge curves of binary composites that show the behaviour of pseudocapacitance and double-layer capacitance. After put into the preceding equation, MXene has a specific capacitance of 194, 140, 142, 137, and 99 F g^−1^ while MoS_2_/MXene has a specific capacitance of 342, 275, 261, 253, and 212 F g^−1^ at 0.4, 1.0, 1.6, 2.0, and 4.0 A g^−1^, respectively.

The GCD curves of MXene and MoS_2_/MXene at 0.4 A g^−1^ are shown in Figure 19b, which unmistakably reveals that the discharge time of MoS_2_/MXene was longer than MXene due to the accelerated ions transfer of MoS_2_ contained into the layers of MXene, which in turn led to improved capacitance. The MoS_2_/MXene composites have a large specific surface area, which greatly minimises the diffusion path and increases interfacial contact, allowing for quick electron transport during the charge/discharge process. The MXene/MoS_2_ composite electrode has the best electrochemical capacitance thanks to its distinctive construction. Table 3 shows recent electrochemical performances for the MXene/MoS_2_ hybrid electrode from various researchers.

## 4. Conclusions and Outlook

Because of its high-power density and long cycle life in a variety of energy storage devices, supercapacitors, which are sometimes referred to as electrochemical capacitors, have garnered a great deal of attention in recent years. MXenes have a wide range of potential applications in the field of supercapacitors due to their superior conductivity, hydrophilicity, as well as their vast range of chemical and structural variety. In the past nine years, there has been a significant rise in the number of studies that investigate the electrochemical characteristics of MXenes and their use as supercapacitor electrode materials. The preparation techniques, composite combinations, and electrodes used during the characterization, on the other hand, were attributed to the current progress made in a solution-based MoS_2_ material supercapacitor and their electrochemical studies. These factors had a significant impact on the results of the electrochemical analyses and the proportion of their cyclic stability. Due to the preparation techniques, composite combinations and electrodes used during the characterisation having a major impact on the output in the electrochemical analyses and the prevalence of them, this condition developed. MXenes are noteworthy in numerous domains, including energy storage, due to its unique electrical and chemical properties. It is expected that the large range of MXenes derivates available in various contexts will provide significant opportunities for future growth. As a result, while creating novel MXenes, it is important to thoroughly investigate their transformation behaviour in a variety of circumstances. It is expected that the large range of MXenes derivates available in various contexts will provide significant opportunities for future growth. According to the findings of the vast majority of studies, the higher electrochemical performance of supercapacitors based on MoS_2_ can be attributed to the shape of MoS_2_ and its composites, as well as to a large specific surface area and rapid charge transfer. Because of their ease of use, broad range of applicability, and relatively excellent electrochemical performance, we can declare for now that hydrothermal methods have been regarded as the most appealing approach to producing MoS_2_ nanosheets due to the fact that they are the most straightforward methods. It is possible to deduce that this strategy is cost-effective because it is generally acceptable. This condition opens the door for its simultaneous application in more than 95% of the published research. 

In the same way as with other types of 2D nanomaterials, increasing the number of active sites requires morphological changes and surface alterations that have a substantial impact. This condition is of utmost significance for manufacturing high-quality MXenes/MoS_2_ with large-scale sheets and nano-scale flaws for high-performance capacitors and batteries. Even though MXenes and MoS_2_ have demonstrated exceptional performance in a variety of applications, particularly in electrochemistry, the physical principles underlying these phenomena still require additional research. It is possible that the restacking propensity of MXene sheets will not be sufficient in achieving its proper electrochemical performance. This condition may have unfavourable effects on developing commercially available devices based on the materials. Another potentially fruitful endeavour is the transformation of MXene and MoS_2_ into marketable products. It is necessary to consider the technological challenges surrounding mass production and process integration to fulfil the prerequisites of an industrial application. For this reason, it is essential to have a solid understanding of the exfoliation mechanism from the most fundamental structure and to investigate the mechanism of the fundamental properties and functionality of MXenes/MoS_2_ in order to scale up the production process in a way that is friendly to the environment and does not incur excessive costs. The development of supercapacitors that are both flexible and wearable should make it possible, in the not-too-distant future, to realise improved storage capacity.

## Figures and Tables

**Figure 1 micromachines-13-01837-f001:**
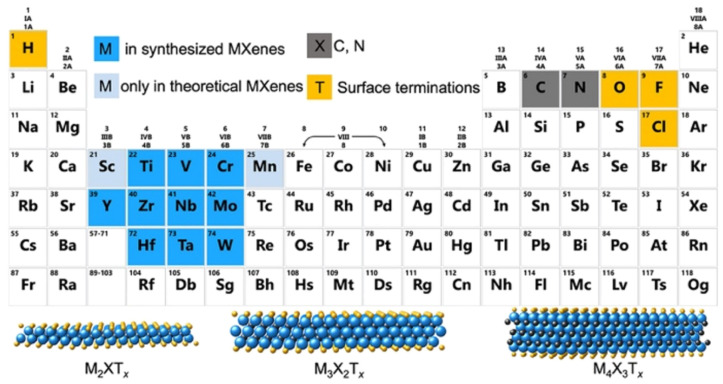
Periodic tables that display the elements necessary to make MXenes. The bottom of the figure contains the schematics for three typical MXene structures. Copyright 2019 ACS Publications.

**Figure 2 micromachines-13-01837-f002:**
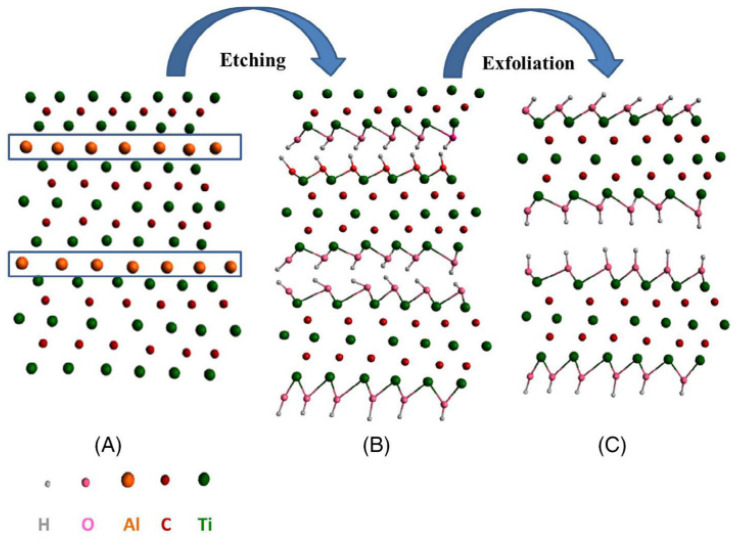
Schematic diagram of the formation of MXene sheets by the exfoliation process of Ti3AlC2. Reproduced with permission from [18]. Copyright 2021 Wiley.

**Figure 3 micromachines-13-01837-f003:**
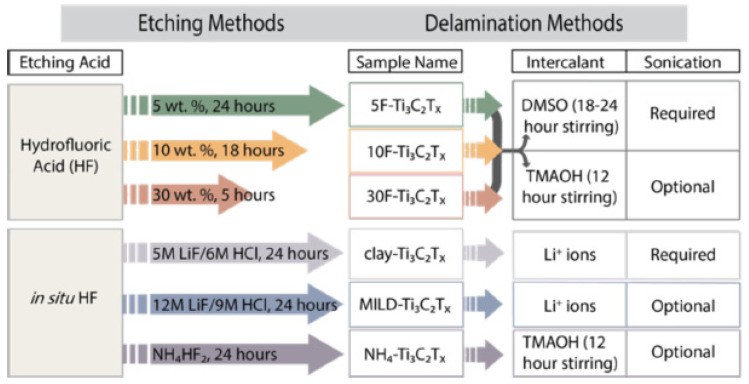
Schematic diagram of employment three distinct concentrations of HF for three separate times, and this will be adequate to synthesise Ti_3_C_2_. Reproduced with permission from [42]. Copyright 2015 Wiley.

**Figure 4 micromachines-13-01837-f004:**
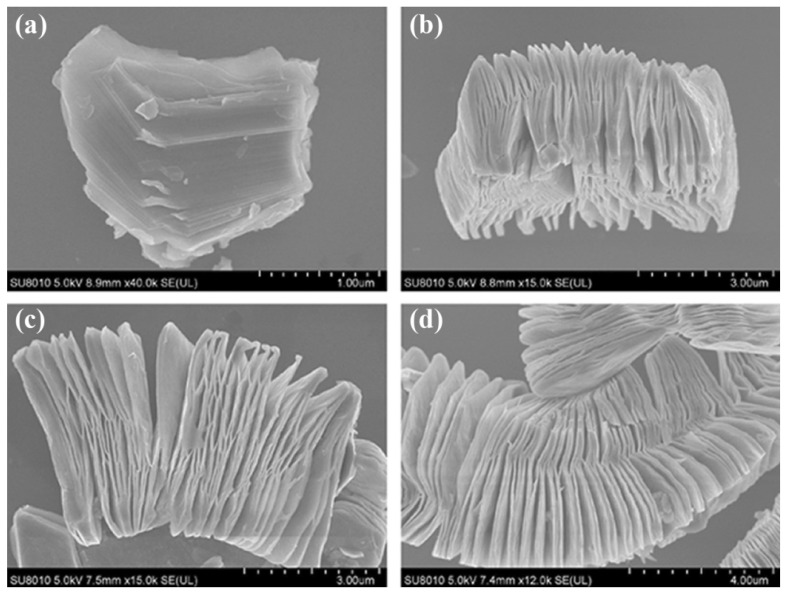
Ti_3_AlC_2_ SEM pictures taken (**a**) before and after HF etching for (**b**–**d**) 2 h, 10 h, and 20 h. Hydrofluoric acid; SEM: scanning electron microscopy. Reproduced with permission from [49]. Copyright 2019 Sage Journals.

**Figure 5 micromachines-13-01837-f005:**
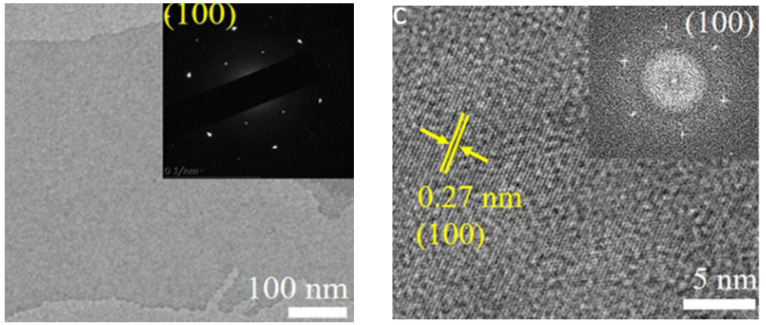
HR-TEM micrograph of the fluoride-free type of method to synthesis MXene. Reproduced with permission from [49]. Copyright 2018 Wiley.

**Figure 6 micromachines-13-01837-f006:**
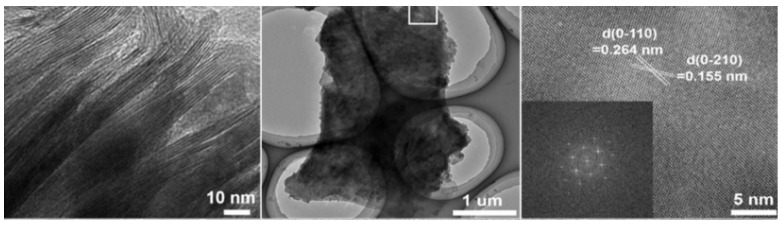
Specific region h-Ti_3_C_2_ nanoflakes in HR-TEM micrograph. Reproduced with permission from [65]. Copyright 2019 Wiley.

**Figure 7 micromachines-13-01837-f007:**
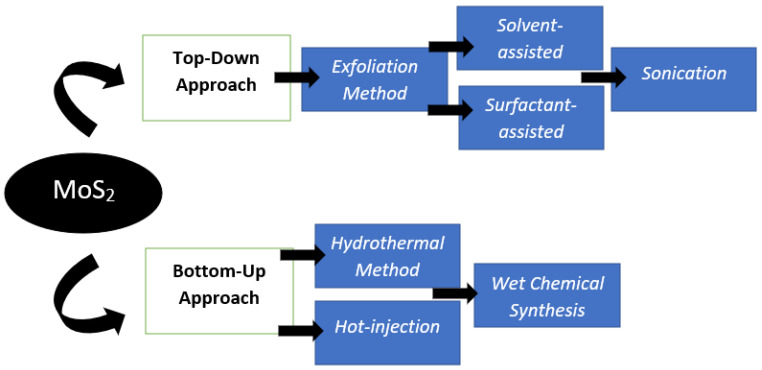
Overview diagram of the MoS_2_ preparation procedures.

**Figure 8 micromachines-13-01837-f008:**
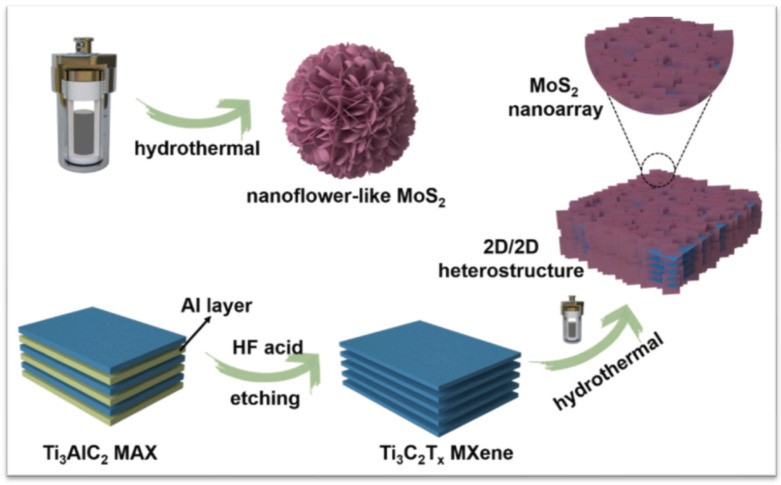
Illustration of the preparation process of MXene-MoS_2_. Reproduced with permission from [85]. Copyright 2021 Elsevier.

**Figure 9 micromachines-13-01837-f009:**
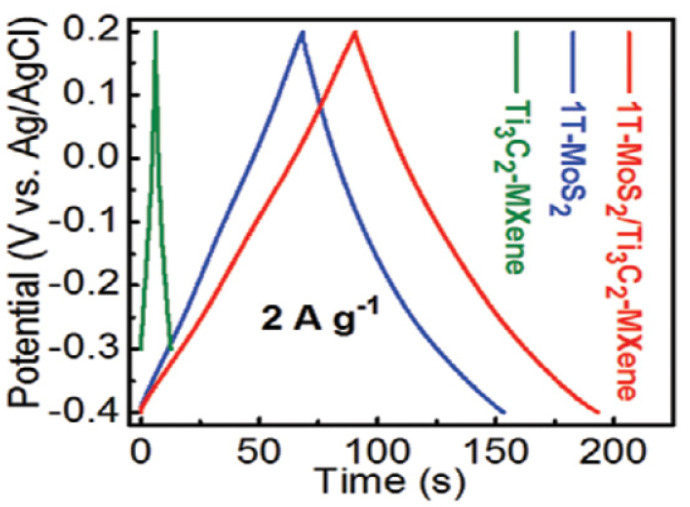
GCD curves 1T-MoS_2_, Ti_3_C_2_ MXene. Reproduced with permission from [88]. Copyright 2022 Elsevier.

**Figure 10 micromachines-13-01837-f010:**
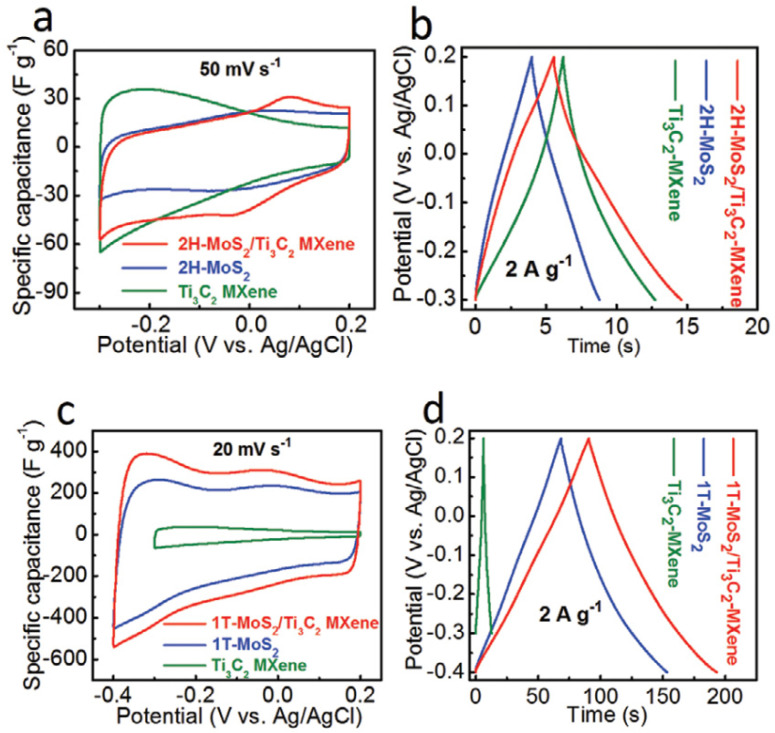
Electrochemical performance of 2H-MoS_2_, 1T-MoS_2_, Ti_3_C_2_ MXene, 2H-MoS_2_/Ti_3_C_2_ MXene, and 1T-MoS_2_/Ti_3_C_2_ MXene electrodes from ref. [88]. (**a**) Specific capacitance of the electrode at 50 mVs^−1^ scan rates (**b**) GCD curves of the electrodes at 2 A g^−1^ current density of 2H-MoS_2_/Ti_3_C_2_ MXene, (**c**) Specific capacitance of the electrode at 20 mVs^−1^ scan rates, (**d**) GCD curves of the electrodes at 2 A g^−1^ current density of 2H-MoS_2_/Ti_3_C_2_ MXene. Copyright 2020 Wiley.

**Figure 11 micromachines-13-01837-f011:**
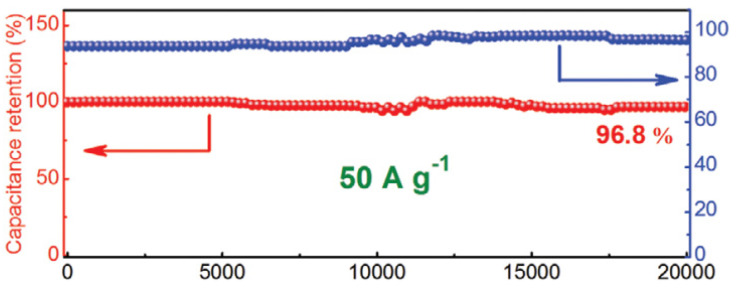
Capacitance retention after 20,000 cycles for 1T-MoS_2_/Ti_3_C_2_ MXene electrode at 50 A g^−1^ from ref. [88]. Copyright 2020 Wiley.

**Figure 12 micromachines-13-01837-f012:**
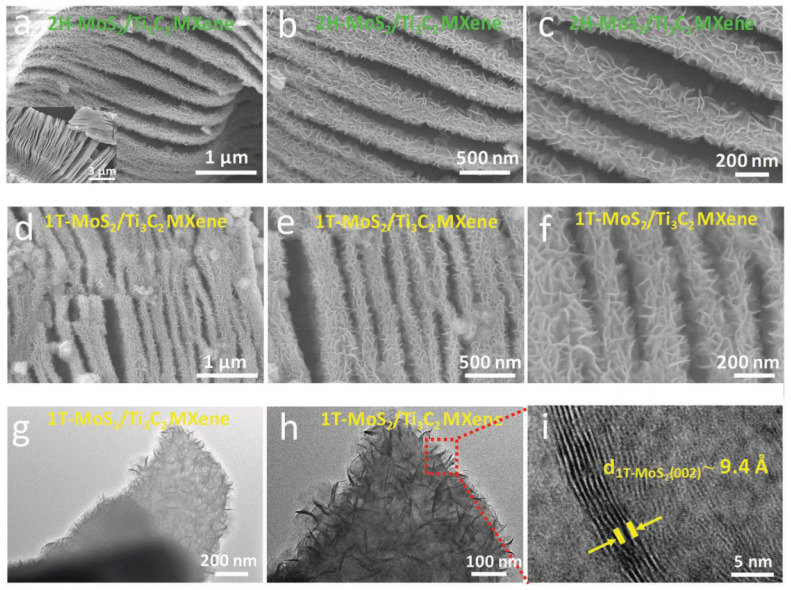
Heterostructures of 2H-MoS_2_/Ti_3_C_2_ and 1T-MoS_2_/Ti_3_C_2_ with different morphologies by using SEM and TEM image. Reproduced with permission from [87]. Copyright 2019 Wiley.

**Figure 13 micromachines-13-01837-f013:**
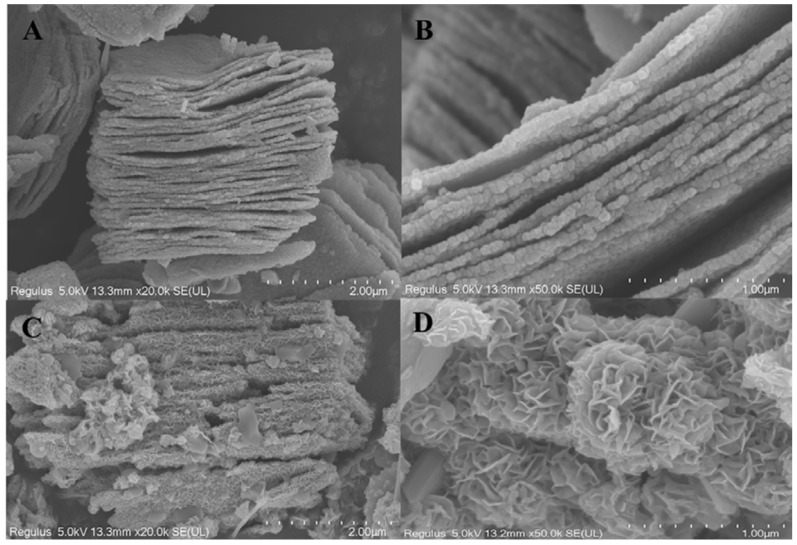
SEM images of pristine Ti3C2 (**A**,**B**) and Ti3C2-MoS_2_ (**C**,**D**). Scale bar = 2 µm and 1 µm. The flower-like particles on layered sheets are observed from the SEM image of Ti_3_C_2_-MoS_2_ from ref. [91]. Copyright 2022 Elsevier.

**Figure 14 micromachines-13-01837-f014:**
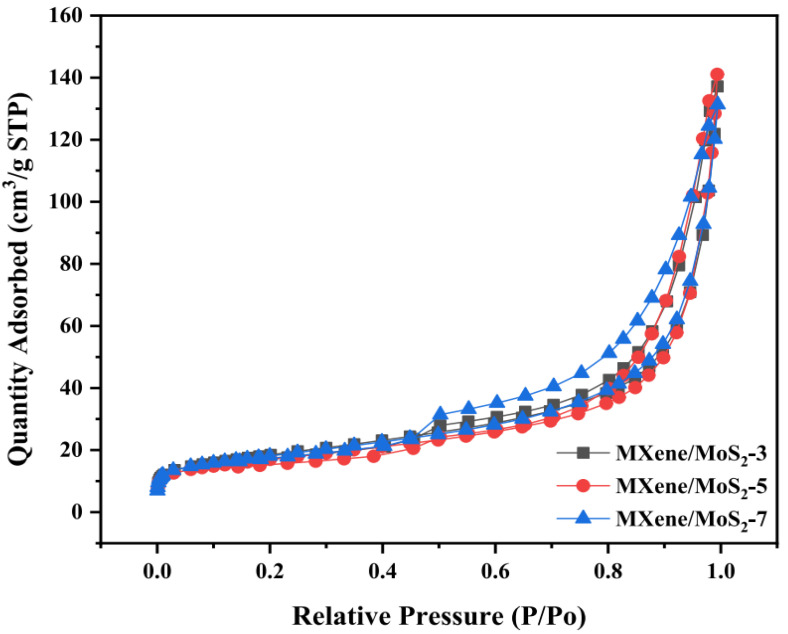
Nitrogen adsorption–desorption isotherms during BET testing. Reproduced with permission from [87]. Copyright 2022 Elsevier.

**Figure 15 micromachines-13-01837-f015:**
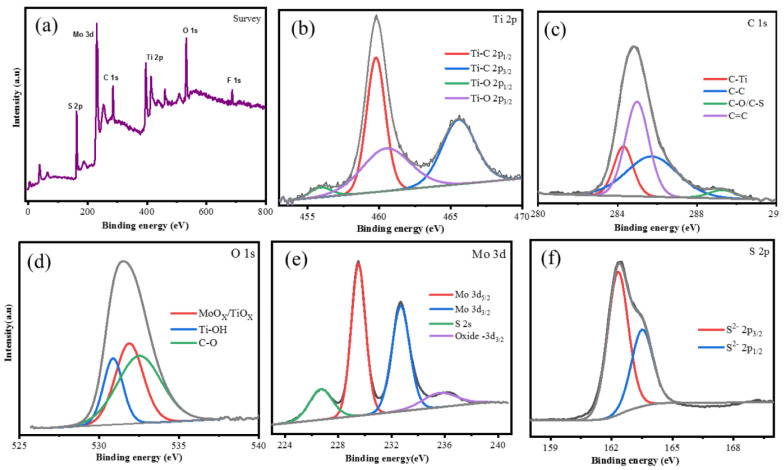
XPS for the (**a**) Survey; (**b**) Ti 2p; (**c**) C 1s; (**d**) O 1s; (**e**) Mo 3d and (**f**) S 2p region for MoS_2_/MXene. Reproduced with permission from [86]. Copyright 2020 Elsevier.

**Figure 16 micromachines-13-01837-f016:**
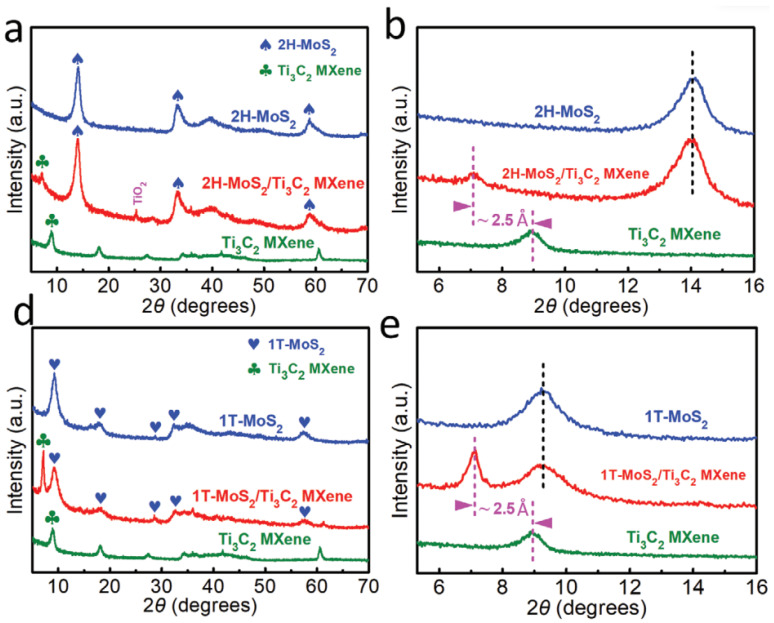
The crystal structure of samples. (**a**) XRD patterns of the 2H-MoS_2_, the Ti_3_C_2_ MXene, and the 2H-MoS_2_/Ti_3_C_2_ MXene heterostructure, (**b**) The magnification of XRD patterns in (**a**), (**d**) XRD patterns of the 1T-MoS_2_, the Ti_3_C_2_ MXene, and the 1T-MoS_2_/Ti_3_C_2_ MXene heterostructure, (**e**) The magnification of XRD patterns in (**d**). Reproduced with permission from [87]. Copyright 2019 Wiley.

**Figure 17 micromachines-13-01837-f017:**
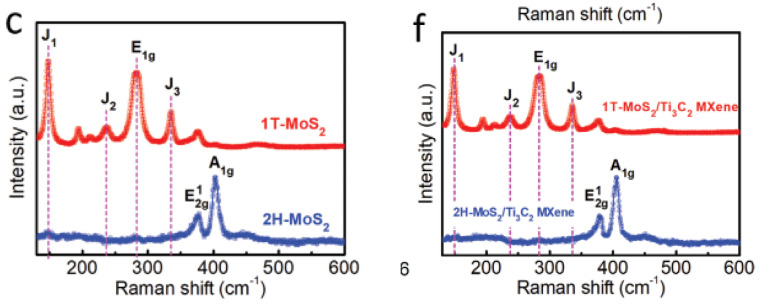
Raman spectra of the 2H-MoS_2_/Ti_3_C_2_ MXene and 1T-MoS_2_/Ti_3_C_2_ MXene. (**c**) Raman spectra of the 2H-MoS_2_ and 1T-MoS_2_, (**f**) Raman spectra of the 2H-MoS_2_/Ti_3_C_2_ MXene and 1T-MoS_2_/Ti_3_C_2_ MXene. Reproduced with permission from [87]. Copyright 2019 Wiley.

**Figure 18 micromachines-13-01837-f018:**
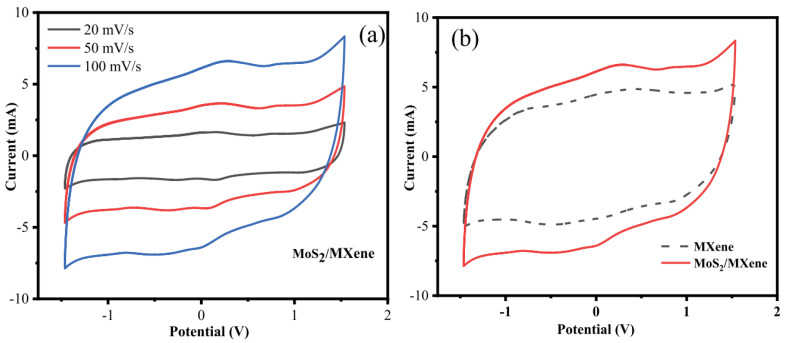
Cyclic voltammetry for MoS_2_/MXene from ref. [87]. Copyright 2020 Elsevier. (**a**) CV curve for MoS_2_/MXene electrode in various scan rates, (**b**) CV curve for MoS_2_/MXene and MXene electrodes in 100 mV s^−1^ scan rates.

**Figure 19 micromachines-13-01837-f019:**
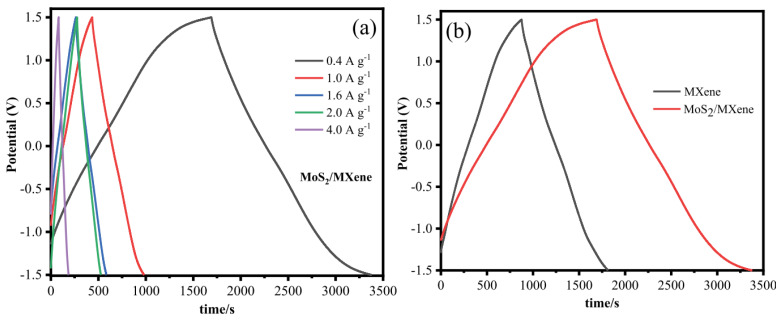
Galavostatic charge–discharge curve of MoS_2_/MXene from ref. [87]. (**a**) GCD curves of MXene/MoS_2_ at at 0.4, 1.0, 1.6, 2.0, and 4.0 A g^−1^ current densities, (**b**) GCD curves of MXene/MoS_2_ and MXene at 0.4 A g^−1^ current densities. Copyright 2020 Elsevier.

**Table 1 micromachines-13-01837-t001:** MXene compositions that have been successfully created in the lab with the general formula of M_n+1_X_n_T_x_. M could be either a single or multiple transition element. Reproduced with permission from [18]. Copyright 2021 Wiley.

	MXenes Are of General Formula M_n,_ Where n = 1–4			
	n = 1	n = 2	n = 3	n = 4
Experimental	Ti_2_C, Ti_2_N, V_2_C, Nb_2_C, MO_2_C, W_2_C, MoN, (Ti,V)_2_C, (Ti,Nb)_2_C, (Mo,V)_2_C, Nb_2_C	Ti_3_C_2_, Ti_3_(C,N)_2_, Hf_3_C_2_, Zr_3_C_2_, (Ti,V)_3_C_2_, (Cr,V)_3_C_2_, (Cr_2_,V)C_2_, (Mos,Ti)C_2_, (Cr_2_,Ti)C_2_, (MO_2_,Sc)C_2_	Ti_4_N_3_, V_4_C_3_, Nb_4_C_3_, Ta_4_C_3_, (Nb,V)_4_C_3_, (Ti,Nb)_4_C_3_, (Nb,Zr)_4_C_3_, (MO_2_Ti_2_)C_3_	(Mo_4_V) C_4_
Theoretical	Sc_2_C, Zr_2_C, Zr_2_N, Hf_2_C, Hf_2_N, V_2_N, Ta_2_C, Cr_2_C, Cr_2_N	Ti_3_N_2_, (Ti_2_Ta) C_2_, (Ti_2_Nb) C_2_, (MO_2_V) C_2_, (Cr_2_Nb) C_2_, (Cr_2_Ta) C_2_, (MO_2_Nb) C_2_, (MO_2_Ta) C_2_	(Ti1Ta_2_) C_3_, (Ti_2_Nb_2_) C_3_, (V_2_Ti_2_C_3_, (V_2_Ta_2_) C_3_, (V_2_Nb_2_) C_3_, (Nb_2_Ta_2_) C_3_, (Cr_2_Ti_2_) C_3_, (Cr_2_V_2_) C_3_, (Cr_2_Nb_2_) C_3_, (Cr_2_Ta_2_) C_3_, (MO_2_V_2_) C_3_, (MO_2_Nb_2_) C_3_, (MO_2_Ta_2_) C_3_	
	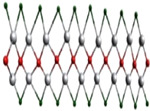	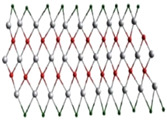	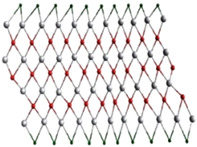	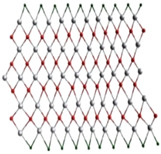

**Table 2 micromachines-13-01837-t002:** Average references of pore size of MXene/MoS_2._ Reproduced with permission from [86]. Copyright 2020 Elsevier.

Samples	BET Surface Area (m^2^/g)	Pore Volume (cm^3^/g)	Pore Size (nm)
MXene/MoS_2_ test 1	65.7671	0.2122	129
MXene/MoS_2_ test 1	60.1815	0.2182	145
MXene/MoS_2_ test 1	64.1727	0.2031	126

**Table 3 micromachines-13-01837-t003:** Recent electrochemical performances for the MXene/MoS_2_ hybrid electrode.

MXene/MoS_2_ Hybrid Electrodes	Electrolyte Used	Specific Capacitance (F g^−1^)	Cyclic Stability
By X. Wang et al. in 2020 [87]	1_M_ KOH	357.4	91.1% after 20,000 long-term cycles
By W. Ding et al. in 2019 [43] [NO_PRINTED_FORM]	1_M_ NaCF_3_SO_3_ in (CH_3_OCH_2_CH_2_)^2^O	224.0	No change in structure after 1 year
By M. Chandran et al. in 2020 [86]	1_M_ H_2_SO_4_	342.00	99% after 10,000 cycles
X. Chen et al. in 2019 [89]	ZnSO_4_ aqueous gel	173.60	98% after 6000 cycles

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
