# Peer review of "Recent Progress of Electrode Architecture for MXene/MoS2 Supercapacitor: Preparation Methods and Characterizations"

_micromachines, 2022, doi:10.3390/mi13111837_

Round 1
Reviewer 1 Report
This review examines the newest developments in MXene/MoS2 supercapacitors, primarily focusing on compiling literature from years of 2017 through 2022. This review also presents an overview of the design (structures), recent developments, and challenges of the emerging electrode materials, with thoughts on how well such materials function electrochemically in supercapacitors. But the discussion is incomplete.
1. The authors have few discussions about the electrochemical properties of MXene/MoS2 in supercapacitors.
2. As a typical two-dimensional material, the layer number and thickness of MXene/MoS2 are very important indications. The authors have few discussions on the thickness and layer number of material by different preparation methods.
Author Response
Dear reviewer, Thank you for your comments
Kindly please see the attachment
Thank you very much

Reviewer 2 Report
The manuscript on Recent Progress of Electrode Architecture for MXene/MoS2 Supercapacitor: Preparation Methods and Characterizations is interesting and but the review is not written well. However, this review article could be improved by adding more recent processes on MXene/MoS2 super capacitive performance and how their architecture helps the improvement of electrochemical properties should be addressed,
I would like to suggest a few things to improve the manuscript,
1. The author should include the MXene stability in an environmental condition and discuss the oxidation and degradation of MXene in the introduction section.
2. It is requested provide and discussion on the formation mechanism of MXene/MoS2 composites.
3. Table 2 does not provide any important information related to the synthesis and the year mentioned is not the first report on that synthesis process. So table 2 may be removed.
4. The author should provide a table to show a summary of MXene/MoS2 composites electrochemical supercapacitor performance including potential window, the electrolyte used, specific capacitance, and cyclic stability.
5. This review is on the recent progress of MXene/MoS2, but only one figure (Figure 7) is provided for the GCD performance. It is suggested to provide a more recent CV, GCD, and cyclic stability performances as coupled figures of composites from available reports. As there are a lot of reports are there on the composites for SC applications
6. Regarding the architecture of the composites, only one figure (Figure 8) has been provided and discussed. I suggest providing more architecture of the composites and how their morphology or architecture helps the improved super capacitive performance should be discussed in detail.
7. The well-known published articles on the MXEne and their composites with good mechanisms and explanations were not referred to. I suggest the authors to refer more articles on the MXene and their composites for better discussions.
8. There is no discussion of flexible architecture. It should be there in the recent progress.
Author Response
Dear reviewer, Thank you for your comments,
Kindly please see the attachment

Reviewer 3 Report
The manuscript reviews the recent development and application of Mxene/MoS2 in supercapacitor as electrodes. The review article first introduced the state of the art, as well as the challenges that researchers face. Then the authors listed the methods for the fabrication/synthesis of Mxene, MoS2, and Mxene/MoS2 stacks. At last, the authors pointed out the future works that need to be done to deal with the current challenges.
Despite the fact that more advanced and comprehensive review articles on the same topic have been published recently, the submitted manuscript focuses more on the comparison of methodologies while the others focus more on applications, therefore, the submitted manuscript still has added value to the scientific society. It can be considered publishing in the Micromachines after minor revisions.
Some remarks are reported in the following:
1. Although ‘MXene’ is already well known among the researchers who work in this field, it is preferred to have an explanation of what it is, especially in a review paper.
2. The numbers in a chemical should be formatted as subscripts. Please go through the manuscript and revise accordingly.
3. It is recommended to include recent review articles, to make this manuscript more comprehensive. Examples: https://doi.org/10.1002/adfm.202110267, https://doi.org/10.1002/aenm.202103867.

Author Response
Dear reviewer,
Thank you for your comments,
kindly please see the attachment

Round 2
Reviewer 1 Report
None
Reviewer 2 Report
The authors have improved the manuscript. The revised manuscript may be considered for publication.